# Effect of Petal Color, Water Status, and Extraction Method on Qualitative Characteristics of *Rosa rugosa* Liqueur

**DOI:** 10.3390/plants11141859

**Published:** 2022-07-15

**Authors:** Giancarlo Fascella, Francesca D’Angiolillo, Michele Massimo Mammano, Giuseppe Granata, Edoardo Napoli

**Affiliations:** 1Research Centre for Plant Protection and Certification (CREA), Council for Agricultural Research and Economics, 90011 Bagheria, Italy; francescadangiolillo@libero.it (F.D.); massimo.mammano@crea.gov.it (M.M.M.); 2Institute of Biomolecular Chemistry (ICB), National Research Council (CNR), Via P. Gaifami 18, 95126 Catania, Italy; giuseppe.granata@icb.cnr.it (G.G.); edoardo.napoli@icb.cnr.it (E.N.)

**Keywords:** rose petals, flower color, bioactive compounds, antioxidant activity, maceration, rose liqueur

## Abstract

Flowers of *Rosa rugosa* Thunb. are a rich source of bioactive compounds with high antioxidant properties and are used for the production of jams, teas, juices, and wines. In the present paper, the petals of *R. rugosa* cv. Alba (white flowers) and Rubra (purple flowers) were evaluated for their morphological and phytochemical characteristics, and for the preparation of an alcoholic liqueur. In particular, the effect of two extraction procedures (conventional and maceration) of fresh and dry petals on the quality of a rose liqueur was determined. As expected, the concentration of the flower’s bioactive compounds was affected by petal water content and by tested cultivars: dry petals showed higher total carotenoids and anthocyanins contents with respect to the fresh ones; cv. Rubra evidenced higher values compared to cv. Alba. As regards the quality of rose liqueur, the two petal extraction procedures did not affect the polyphenol content and higher values were recorded only on dry petals with respect to the fresh ones and, in particular, on those from cv. Rubra. The liqueur’s flavonoid content was influenced by the petal extraction method, water content, and color as higher values were recorded on rose liqueur prepared after the maceration of cv. Rubra dry petals whereas lower values were observed on alcoholic drinks prepared after the conventional extraction of cv. Alba fresh petals. Our study shows that *R. rugosa* petals have a fair amount of secondary metabolites with antioxidant activity, making them suitable for use in the beverage industry.

## 1. Introduction

Recently, the interest in functional foods and beverages prepared from fresh and local products has increased [1]. Many research programs deal with the valorization of some traditional ingredients and their potential health benefits. Roses are recognized as edible flowers and have been traditionally used as food components, both in the fresh form and in processed products, such as sweets and drinks [2,3]. Some authors [4,5] include among the edible roses those species and cultivars whose petals are not astringent or bitter and with a texture suitable for fresh and dry consumption. *Rosa rugosa* Thunb. (also known as rugose rose, wrinkled rose, or Japanese rose) is a 1.5–2 m tall shrub native to eastern Asia, China, Korea, and Japan, cultivated for ornamental purposes and appreciated for fragrant colorful flowers in many parts of world [6,7]. This species is very adaptable, showing heat and drought tolerance as well as cold hardiness and can be grown in full sun and partial shade [8]. In addition, it shows a high resistance to main pests and diseases. *R. rugosa* is widely cultivated for the production of flowers, fruits, and leaves, which represent a source of bioactive compounds with antioxidant potential [6,9]. Indeed, this species has been used in Chinese folk medicine and the food industry for centuries. Different plant organs have been traditionally used against diarrhea, gastroenteritis, hepatitis, injuries, and blood circulation disorders [10]. The petals are characterized by a high concentration of different biologically active substances, such as anthocyanins, flavonoids, and phenolic acids, with a high antioxidant activity and consequently with protective effects against many human diseases such as intestinal and pathogenic bacteria and prostate cancer [11,12]. For these reasons, *R. rugosa* petals are commonly used as spices and for the preparation of healthy foods (jams, jellies, leavens, and syrups) and drinks (teas, juices, and wines) [9,13,14]. Among these latter products, liqueurs and alcoholic beverages obtained from hand-picked rose petals may show positive effects on human health, if moderately consumed [1]. These alcoholic (15–40%) products may differ in color, taste, scent, and other sensorial parameters depending on the rose cultivars used and the color of their flowers [15]. It is known that the intense color of rose flowers is linked to several classes of phenolic compounds, suggesting their suitability for the preparation of aromatic liqueurs [16]. Traditionally, rose liqueur is prepared by drenching petals in alcohol as immersion may gradually extract the bioactive compounds into the liquid. Phenolics are the major antioxidant molecules in these products, and their extraction is related to pre-treatment and drenching conditions [17]. Among the most used techniques, fragrant petals are drenched in ethanol, left at room temperature in the dark for a couple of weeks, with a final addition of sugar. Nevertheless, the available scientific literature regarding the health-promoting potential of flowers from *R. rugosa* genotypes is still inadequate. In particular, a few studies about the effect of *R. rugosa* cultivars on petal compounds content and on quality of petal-derived beverages have been published [18,19]. Therefore, the purpose of current study was (1) to evaluate the secondary metabolites content of fresh and dry petals of two *R. rugosa* cultivars with different flower colors and (2) to compare the qualitative characteristics of alcoholic liqueurs prepared from the above-mentioned petals using two different extraction methods.

## 2. Materials and Methods

### 2.1. Plant Materials and Growth Conditions

Cutting-propagated plants of *Rosa rugosa* Thunb. cv. Alba (white corolla color) and cv. Rubra (pink corolla color) grown in open air at the experimental farm of the Research Centre for Plant Protection and Certification of Bagheria (38°5′ N, 13°30′ E, 23 m above sea level), north-west Sicily, were used for the present study (Figure 1). Fifty fresh flowers were randomly selected in May 2019 (max/min temperature 25.6/15.4 °C) from 5-year-old bushes of both cultivars (10 flowers/plant) at full opening development stage. The bushes were planted on a Mediterranean red soil, in single rows at 10 plants m^−2^ density, with summer drip irrigation and no chemical spraying. After collection, the petals were packed in paper bags and transferred on ice to the laboratory where they were immediately analyzed.

### 2.2. Morphological Description and Color Parameters

For each of the two considered cultivars, main morphological characteristics of rose petals (length, width, length to width ratio, and fresh and dry weight) were described. Fifty petals per cultivars were randomly picked up from different levels of the canopy, directly measured and weighed. Petal length (L) was measured with a digital caliper from the top of the petal to its base; petal width (W) was measured at the widest part of the petal, perpendicular to its length. Petal fresh weight (FW) was determined immediately after harvest. Petal dry weight (DW) was recorded after air-drying in the laboratory at a room temperature of 25 °C for 5 days; then, petals were dried in a heated chamber (45 °C) until a constant weight was reached. Petal water content (%) was calculated according to the formula: [(FW–DW)/FW × 100].

The color coordinates of fresh and dry petals of both cultivars were measured in the middle of each petal (fifty for each treatment) with a portable colorimeter (CR-10 Chroma; Minolta, Osaka, Japan) that had been previously adjusted with a white calibration plate. According to the CIE L° a° b° color representation system, the L° coordinate corresponds to a dark–bright scale and indicates the lightness, with values ranging from 0 (black) to 100 (white). Color coordinates a° and b° range from −60 to +60, where negative and positive a* values indicate greenish and reddish colors, respectively, and negative and positive b* values correspond to bluish and yellowish colors, respectively. Chroma (C*) represents the brightness and was calculated according to the equation: (a°^2^ + b°^2^)^1/2^. Moreover, hue angle, expressed in degrees, was calculated according to the formula: h° = tan^−1^(b°/a°) with values ranging from 0 to 360 where 0° or 360° = red-purple, 90° = yellow, 180° = green, 270° = blue.

### 2.3. Determination of Petals Total Carotenoids Content

The carotenoids content of fresh and dry petals of the two R. rugosa cultivars were determined using the method described by D’Angiolillo et al. [20]. The petals (500 mg of fresh and 100 mg of dry material) were homogenized in 25 mL of commercial ethyl alcohol (95% vol.). After 24 h of incubation at 4 °C in the dark, the absorbance of the extracts was determined at 470 nm by means of a UV/Vis spectrophotometer (Beckman DU 530, GMI, Ramsey, MN, USA). The total carotenoids content (expressed as mg g*^−^*^1^) was estimated based on the absorbance values as per standard equations of β-carotene compound.

### 2.4. Determination of Petals Total Anthocyanins Content

Determination of the total anthocyanin content of petals was done according to D’Angiolillo et al. [20]. The petals (200 mg of fresh and 50 mg of dry material) were extracted in a volume of EtOH/HCl (*v*/*v* 99/1%) with the addition of 2/3 volume of distilled water. The extracts samples were centrifuged at 12,000 rpm for 20 min at room temperature. The absorbance of anthocyanins was determined spectrophotometrically at 535 nm. Total anthocyanins content (expressed as mg/100 g) was calculated basing on the standard curve prepared using cyanidin chloride at different concentrations (1–200 µg/mL).

### 2.5. Petals Extraction and Liqueur Preparation

The fresh and dried petals collected from the two R. rugosa cultivars were used to prepare two different extracts to be successively used for the preparation of a traditional liquor (Figure 2). For the conventional extraction (CON) the samples (0.5 g of fresh and 0.1 g of dry petals) were pulverized and homogenized in a mortar with 6 mL of 70% (*v*/*v*) of commercial ethyl alcohol (95% vol.) to facilitate the extraction. After 30 *min* of incubation on ice, the extracts were centrifuged at 12,000 rpm for 10 min at room temperature. For each sample, the supernatant (ethyl alcohol extract) was collected and analyzed. While, for the maceration extraction (MAC) the samples (1.5 g of fresh and 0.3 g of dry petals) were left to soak in 20 mL of 70% (*v*/*v*) of commercial ethyl alcohol (95% vol.) for 14 days in the dark and at the room temperature (Figure 3). All of the extractions collected (Alba CONfresh; Alba CONdry; Rubra CONfresh; Rubra CONdry and Alba MACfresh; Alba MACdry; Rubra MACfresh; Rubra MACdry) were performed in three technical repetitions and were used for the determination of polyphenols, flavonoids, and the antioxidant activity.

### 2.6. Spectrophotometric Determination of the Liqueurs Total Polyphenol Content

Total polyphenol contents of different typologies of petals extract were determined by the Folin*–*Ciocalteu (FC) colorimetric method with slight modification [21]. After dilution in 70% ethanol (1:1 *v*/*v*) of the flower extracts, a 0.005 mL aliquot of diluted extract was mixed with FC’s reagent (0.5 mL) and kept for 5 min at room temperature. Then, 0.45 mL of a 7.5% (*w*/*v*) Na_2_CO_3_ solution was added to the mix and incubated for 2 h in the dark at 20 °C. Absorbance at 765 nm was read on a spectrophotometer (Beckman DU 530, GMI, Ramsey, MN, USA) after 2 h at room temperature. Chlorogenic acid (CA) was used to calculate the standard curve and total polyphenol contents were expressed as mg of CA equivalents per g of extract.

### 2.7. Spectrophotometric Determination of the Liqueurs Total Flavonoid Content

The total flavonoids content of the petals extract was determined according to the colorimetric method followed by Fascella et al. [21]. An aliquot of 0.35 mL of distilled water and 0.075 mL of 5% (*w*/*v*) sodium nitrite were added to 5 µL of extract and, after 5 min of incubation, 0.075 mL of 10% (*w*/*v*) of aluminum trichloride hexahydrate was added to the mix. After 5 min, 0.500 mL of 1M sodium hydroxide was added. The samples were incubated for 15 min at room temperature and the absorbance was determined spectrophotometrically at 415 nm. Quercetin (Q) was used to calculate the standard curve and total flavonoids were expressed as mg of Q equivalents per g of extract.

### 2.8. HPLC-DAD Liqueur’s Anthocyanins Content

Quantitative analyses of anthocyanins were carried out as previously described [22] on an Ultimate3000 ‘UHPLC focused’ instrument equipped with a photodiode array detector (Thermo Scientific, Rome, Italy). Chromatographic runs were all performed using a reversed-phase column (Gemini C_18_, 250 × 4.6 mm, 5 μm particle size, Phenomenex, Bologna, Italy). An aliquot of 20 µL of each liqueurs sample were eluted with the following gradient of B (formic acid, 2.5% solution in acetonitrile) in A (2.5% solution of formic acid in water): 0 min: 10% B; 20 min: 35% B; and 25 min: 10% B. The solvent flow rate was 1 mL/min. DAD analyses were carried out in the range between 700 and 200 nm, registering the chromatograms at 280, 330, 350, and 520 nm. Quantifications were carried out at 520 nm. Cyanidin-3-O-glucoside (R^2^ = 0.9931) and malvidin-3-O-glucoside (R^2^ = 0.9907) were used as analytical standards. Analyses were always carried out in triplicate.

### 2.9. Evaluation of Diphenyl Picrylhydrazyl (DPPH) Radical Scavenging Activity

The radical scavenging activity of the liqueurs was evaluated by DPPH assay [23,24,25]. Briefly, aliquots of each liqueur were previously diluted with methanol (1:5 *v*/*v* ratio), then, suitable amounts of these solutions were mixed with 3 mL of 0.1 mM DPPH in methanol to obtain mixtures containing a liqueur concentration ranging from 0 to approximately 1.5 mg/mL (except to Rubra-CONdry, 0–1.19 mg/mL). The solutions were stirred at 25 °C in the dark for 30 min and the absorbance of each at λ = 517 nm was recorded on a UV–VIS spectrophotometer (8453 UV–Visible spectrophotometer; Agilent Technologies, Santa Clara, CA, USA), so as to obtain the values of radical scavenging activity (*RSA*(%)) by using the following formula:$$RSA\left(\%\right)=\frac{A_{0}-A_{s}}{A_{0}}\times 100$$
where *A_0_* and *A_s_* are the absorbance values of the DPPH in absence or in presence of sample, respectively. The SC*_50_* values, defined as the amount of liqueur necessary to scavenge half of the initial DPPH concentration, were calculated by plotting the *RSA*(%) versus liqueur concentration (R^2^ ≥ 0.994).

### 2.10. Statistical Analysis

The obtained data were subjected to statistical analysis performed with PAST version 4.03 (Natural History Museum, University of Oslo, Oslo, Norway) software package. A one-way analysis of variance (ANOVA) was applied to the data related to the morphological characteristics, color parameters, total anthocyanins, and total carotenoids of fresh and dry petals. A two-way (2 petal extractions × 2 cultivars) ANOVA was applied to data related to the total polyphenols total flavonoids and anthocyanins of rose liqueurs. Significant differences (*p* < 0.05 and 0.01) in all the considered parameters were tested using Duncan’s multiple-range tests.

## 3. Results and Discussion

### 3.1. Petals Morphology and Color Measurement

The two R. rugosa cultivars tested differed on some morphological characteristics of the flower petals. In particular, the highest petal length and width were recorded on cv. Alba whereas the highest L/W ratio was measured on cv. Rubra (Table 1). No significant differences between cultivars were observed on fresh and dry weight of the petals. As regards the color coordinates, petal lightness (L) was affected by the studied cultivars but not by the petal water content, with higher values (as expected with white flowers) recorded with cv. Alba (Table 2). Significant differences on petal redness/greenness (a°) and on yellowness/blueness (b°) were recorded both between the two cultivars and between fresh and dry petals: particularly, higher a° values were measured on petals from cv. Rubra (purple flowers) and lower measurements on those from cv. Alba. Schmitzer et al. [16] reported an increase in a°, which is associated with the amount of red coloration in the petals, in 72 h air-drying petals of a light pink cultivar. The same authors also referred that b* values were higher in cultivars with white petals and lower in pink-flowered cultivars. Zawiślak and Michalczyk [14], in a study on the effect of sucrose addition and storage length of minimally processed products from R. rugosa petals, measured a b° value of −10.7 on fresh petals. The chroma (C°) was significantly influenced by the cultivars, with higher values recorded on cv. Rubra, but not by water content of the petals (Table 2). The hue angle (h°), which defines the basic color of a sample, was affected by the rose cultivars as well as by the petal water content as higher values were observed on petals from cv. Rubra and on dry petals. An increase in h° was observed by Schmitzer et al. [16] in dry petals of pink colored cultivar after 48 h or 72 h air-exposure. Li et al. [7] reported similar values of b° (−12.64) and h° (337.7) in petals of R. rugosa cv. ‘Hunchun’ with strong reddish-purple flowers.

### 3.2. Petal’s Total Carotenoid and Total Anthocyanin Content

The total carotenoid and anthocyanin content in the extracts from fresh and dry petals of the two cultivars of *Rosa rugosa* are reported in Table 3. Total carotenoids were higher in fresh petals of cv. Rubra, as expected from a cultivar with reddish color petals, with respect to those of cv. Alba (1.04 and 0.61 mg/g, respectively). The carotenoid content of dry petals of cv. Rubra doubled up than that from cv. Alba (4.41 and 2.14 mg/g). Zawiślak and Michalczyk [14] reported that *R. rugosa* petals contained β-carotene. Nowak et al. [9], when investigating *R. rugosa* petals composition, determined the presence of carotenoids in a concentration of 1.39 µg g^−1^. Olech et al. [10] found a high carotenoid content in *R. rugosa* flowers when used as raw plant material. Carotenoids, as sources of provitamin A, are important antioxidant rose constituents protecting against cardiovascular and inflammatory diseases. Anthocyanins were detected in both analyzed cultivars, including that with white flowers. Fresh petals of cv. Rubra showed a higher total anthocyanins content, with respect to those of cv. Alba (24.05 vs. 1.22 mg/100 g, respectively); in the dry petals, the anthocyanins content was higher compared to that from the fresh ones, and the difference between the cultivars is also very evident (60.81 and 9.34 mg/100 g for cv. Rubra and Alba, respectively) (Table 3). As well as expected for carotenoids content, fresh and dry petals from cv. Rubra were characterized by a higher anthocyanins content than that from cv. Alba petals, as shown also by a plethora of papers indicating that petal color intensity is mostly determined by the concentration of accumulated anthocyanins [7]. The content of cyanidin 3−5-di-O-glucoside, which is the predominant anthocyanin of Japanese rose petals, determined by Cendrowski et al. [11] in petals of *R. rugosa* was very similar to the total anthocyanins content measured in our experiment on fresh petals of cv. Rubra (Table 3). Sparinska and Rostok [26] reported that the quantity of anthocyanins measured in the petals of *R. Rugosa* Latvian varieties ranged from 7.84 to 0.26 mg/100g. It is known that anthocyanins play a protective role for human health as they show anti-inflammatory activities and are able to prevent edema [11]. Moreover, by inhibiting free radicals, they may prevent lipid peroxidation and are important in the prevention of colon cancer.

A significant correlation (r^2^ = 0.85) between color coordinate a° and total anthocyanin content of fresh and dry petals of cv. Rubra cultivars was obtained (Figure 4); this outcome confirms the close relationship between the red color of plant tissues and their relative anthocyanin concentration, as reported by other authors [7,16].

### 3.3. Total Polyphenol Content of Liqueurs from Different Petal Extractions

The qualitative characteristics of rose liqueurs were affected by the tested cultivar, the petal water content, and by the extraction procedure. Total polyphenols were higher in liqueurs prepared with dry petals of cv. Rubra, with respect to those from cv. Alba as a mean of the results obtained with the two preparation methods (175.6 and 84.4 mg/g DW, respectively); lower values were recorded in liqueurs prepared with fresh petals of both cultivars (29.7 and 15.0 mg/g FW for cv. Rubra and cv. Alba, respectively) (Table 4). As regards the extraction technique, a higher polyphenols content was measured in liqueurs obtained after conventional extraction of dry petals compared to that from liqueurs prepared after maceration (138.2 and 121.8 mg/g DW, respectively); no significant difference on total polyphenols (22.4 mg/g FW, on average) was recorded in fresh petals treated with the two extraction procedures (Table 4).

Cendrowski et al. [18] reported that the content of polyphenolic compounds in liqueurs from *R. rugosa* petals macerated with 65% ethanol was 95.1 mg/cm^3^, confirming that *R. rugosa* petals, being rich in phenolic compounds, are valuable raw materials as a source of bioactive compounds for the production of functional foods and pro-health preparations. These same authors referred that acidified aqueous ethanol is a more effective solvent than aqueous ethanol at the same concentration for polyphenols extraction from the petals, but that maceration still remains the most common extraction technique for natural products. Actually, traditional maceration is yet suitable for the food industry due to the presence of aqueous ethanol, a common bio-solvent, easily available in high purity and completely biodegradable. Cendrowski et al. [18] stated that the most critical component to produce rose liqueurs is the concentration of alcohol used for the maceration of the rose extract. Shen et al. [17] reported that drying is a good way for preserving rose petals in the process for making of rose liqueur: the best quality was found by soaking dried rose petals in a base spirit containing 30–35% ethanol at a 1:10 (*w/w*) ratio. The same authors observed that a 2-day soaking time seems to be adequate for the preparation of rose liqueur and the phenolic compounds content in the soaking process ranged between 2300 and 2500 ppm. Olech and Nowak [27] reported that the quantity of phenols determined in extracts from *R. rugosa* petals are significantly affected by the extraction conditions and by the used extractant. Actually, these latter authors observed significant differences in total phenolic content when acetone, ethanol, or methanol were used as solvents and a mean value of 139.60 μg/mg of dry extract (corresponding to 74.1 mg/g of raw material) was recorded for a dry extract with 70% ethanol at ambient temperature. Authors such as Vinokur et al. [2] suggested that the polyphenols content in rose petals is deeply affected by environmental factors (altitude, climate, soil substrate, etc.), but considering that our liqueurs were obtained from roses cultivated in the same location and under the same growing technique, the unique difference on total polyphenols may be related to the used cultivars (genotype) as well as to the petals’ water content and the extraction procedures. Schmitzer et al. [1] reported that phenolic profiles of rose liqueurs were cultivar-specific and different patterns in phenolic composition were observed with the diverse extraction procedures. Phenolics in fruit-based alcoholic beverages may potentially reduce the harmful (pro-oxidative) effects of alcohol when liqueurs are consumed in moderate amounts. Hence, it is important to detect procedures that produce liqueurs with a higher phenolics content. Sokół-Łętowska et al. [28] measured a large amount of phenolic compounds, approximately 140 mg/100 mL, in the liqueurs of black rose (*R. spinosissima*). Ozsoy et al. [29], when assessing the total phenolics from flower ethanolic extracts of *R. horrida*, confirmed that rose petals are rich sources of polyphenols: flowers contained 28.7 ± 0.12 mg/g DW of polyphenols and 12.1 ± 0.14 mg/DW of flavonoids. Cendrowski et al. [18] reported that each part of the *R. rugosa* plant contains large amounts of phenolic constituents; in particular, the petals are rich in polyphenols with antioxidant properties, and, consequently, with potential medicinal uses. Liu et al. [30] observed that *R. rugosa* polyphenol-enriched extract reduced blood glucose in type 2 diabetic rats through an enhancement of insulin sensitivity. Ng et al. [31] reported that some phenolic compounds extracted from *R. rugosa* petals inhibited lipid peroxidation.

Our results summarized in Table 4 confirm what is reported above. As expected, liqueur obtained with Rubra cultivar petals contain a higher amount of polyphenols, flavonoids, and anthocyanins. Infusions with dry plant material seems to be more efficient in terms of extraction yields than fresh material independently by the way (conventional or maceration) used. Instead, the two extraction methods seem to have an effect on the quantities of polyphenols and flavonoids in the liqueurs. The conventional method increases the amount of total polyphenols, resulting in a less effective extraction of flavonoids than maceration. This may be due to the different contact times between the solvent and the plant matrix. The shorter contact time of the conventional method could favor the greater solubilization of small phenolic molecules such as acids. During the 14 days of maceration, on the other hand, even larger molecules such as flavonoids and anthocyanins could be solubilized more easily.

Comparing the two preparation methods, dry raw material provides higher amounts of bioactive molecules in the final product than fresh raw material in all proposed treatments (Table 4). This confirms that drying is a good way for preserving rose petals and for inhibiting the enzymatic degradation of anthocyanins in the making of rose liqueur [17]. According to the same authors, the anthocyanins extraction rate was initially faster from fresh petals, likely for the time necessary for the solvent to flow in the shrunken cells of dried petals, whereas, in the latter stage, a higher concentration of anthocyanins was extracted from dried petals due to the dilution effect by the cell sap from fresh petals. Shen et al. [17] also reported that the total anthocyanin content reached the highest value after the 2 day-soaking of petals in the ethanolic solution, and then decreased probably due to the continuation of polymerization and degradation reactions of anthocyanin in the solution.

The anthocyanin content in liqueurs decreases when compared to that from the starting petals. In the white-pigmented cv. Alba, they are below the detection limit of the analytical method. Even the purple-pigmented cv. Rubra, both with the conventional method and with the maceration method, shows a loss of anthocyanins during the preparation of the liqueur. Maceration is a more effective method than the conventional method in extracting anthocyanins from the petals, as shown in Table 4 (13.5 and 2.8 mg/g for maceration and conventional method, respectively). Schmitzer et al. [1] reported that the content of individual and total anthocyanins was dependent on the method of extraction: the highest total anthocyanins concentration was measured in liqueurs made with air-dried intense red petals soaked in 97% and 50% ethanol; however, the ethanolic extraction of air-dried petals gave liqueurs with the highest levels of total anthocyanins, irrespective of the used cultivar. Vinokur et al. [2] reported that the total anthocyanin content in rose teas was correlated with the petal color, with relatively high values in those prepared from the red-flowered cultivars. Among the different anthocyanins detected in rose liqueurs from cv. Rubra, cyanidin 3-*O*-glucoside was the predominant pigment, both in those obtained from fresh and dry petals (Table 4).

### 3.4. Total Flavonoid Content of Liqueurs from Different Petal Extractions

The total flavonoid content was higher in liqueurs prepared with dry petals of cv. Rubra, with respect to those from cv. Alba (89.8 and 52.6 mg/g DW, respectively); lower contents were recorded in liqueurs prepared with fresh petals of the two tested cultivars (20.4 and 10.0 mg/g FW for cv. Rubra and cv. Alba, respectively) (Table 4). With regard to the extraction procedure, higher flavonoids were measured in liqueurs obtained after the maceration of dry petals compared to those from liqueurs prepared after conventional extraction (86.8 and 55.5 mg/g DW, respectively); this difference between the two extraction procedures is also evident in liqueurs made with fresh petals (20.7 and 9.8 mg/g FW for maceration and conventional extraction, respectively) (Table 4). The effect of the cultivars on the quality of drinks prepared with rose petals was previously reported [2,15], as caffeine-free beverages from different rose varieties significantly differed in their sensory properties that are determined by the presence of specific taste- and aroma-creating compounds. Pires et al. [15], when quantifying the phenolic compounds present in rose dry petals, determined that the sum of the flavonoids was 9.18 and 4.24 mg/g DW for hydro-methanolic extracts and infusions, respectively. The best quality performance of dry petals is in line with Shen et al. [17], who observed that dehydration, by inactivating the anthocyanin-degrading enzyme and the browning enzymes in the petals, improves the color quality of rose liqueur (compared to that from fresh petals) and is also an efficient way to preserve rose petals as a semi-product. Such as for the total polyphenols, the flavonoids concentration may be influenced by the extraction conditions and by the used extractant as Olech and Nowak [27] reported that total flavonoids in *R. rugosa* petal extracts was 1.94 μg/mg of dry extract when 70% ethanol at 22 °C was used as a solvent. Cendrowski et al. [11] reported that the most important characteristic of flavonoids is their antioxidant activity, which evidences many pharmacological applications and, especially, the tumor growth inhibition. Flavonoids also play an important role in the prevention of cardiovascular diseases. These compounds are also able to act against pathogenic microorganisms by selectively inhibiting the proliferation of many viruses. Beverages from fruits and vegetables are the major sources of flavonoids whose consumption has been associated with both vascular and cognitive health benefits during a lifetime [32]. The determination coefficient, r^2^, between total polyphenol and total flavonoid content (Figure 5) was 0.77 (significant at the 5% level). This positive and significant correlation is important, from a nutraceutical point of view, because they are strictly connected to the high antioxidant activity of these compound groups with consequent beneficial effects on human health.

The combination of health benefits with documented applicability of *R. rugosa* petals in liquor-making activities may increase the opportunity to use the flowers of this rose species for the preparation of alcoholic functional beverages with high antioxidant properties.

### 3.5. Evaluation of DPPH Radical Scavenging Activity

To compare the antioxidant activity of the various liqueurs, the ability of antioxidants as polyphenols to scavenge the DPPH radical was exploited. Table 5 reports the estimated values of SC_50_ relative to the DPPH scavenging activity of each liqueur. The Rubra-CONdry sample was the most effective liqueur against DDPH (the sample with the lowest SC_50_). As expected, all Rubra extracts were more active than those of Alba due to the higher concentrations of phenolic compounds (in general) and to the presence of anthocyanins (in particular). The data also show that by comparing the methods of preparation of liqueurs (conventional and maceration) for the same cultivars and the same methods of drying the petals, the samples obtained with the conventional method have a greater antioxidant activity with the exception of the Alba-MACdry and Alba-CONdry samples where this trend is reversed (Table 5). If, on the other hand, the antioxidant activity data of the different drying methods of the petals are compared for the same cultivars and the same methods of preparation of the liqueurs, this is higher for the samples prepared with dried petals except for the samples made with Alba petals and the conventional method.

Figure 6 shows the antiradical Power (ARP = 1/SC_50_) of the various rose liqueurs to clearly display their activity scale. Vinokur et al. [2], besides observing differences in antioxidant activity on rose tea prepared from petals of different cultivars, reported that its radical-scavenging activity is mainly linked to the high concentration of phenolic compounds, in particular to the free gallic acid content. Shameh [33] reported that the DPPH antioxidant activity of petals from different rose species varied with the genotype.

## 4. Conclusions

The results of our studies conducted on the extracts and liqueurs from petals of two cultivars of *Rosa rugosa* (Alba and Rubra) show that these plant-based matrices have a fair amount of secondary metabolites with antioxidant activity, especially flavonoids and anthocyanins. This makes them suitable for use in the food and beverage industry. In this study, taking as a reference the antioxidant activity of the finished product (liqueur), two methods of preparation (conventional and maceration) were also compared. Our results seem to indicate that the conventional method using dried petals leads to a product with a higher concentration of polyphenols and, therefore, with a higher antioxidant activity. Obviously, further studies are required in order to detect how other liqueur’s bio-components (single polyphenols and flavonoids) are affected by the rose cultivars and by the extraction method.

## Figures and Tables

**Figure 1 plants-11-01859-f001:**
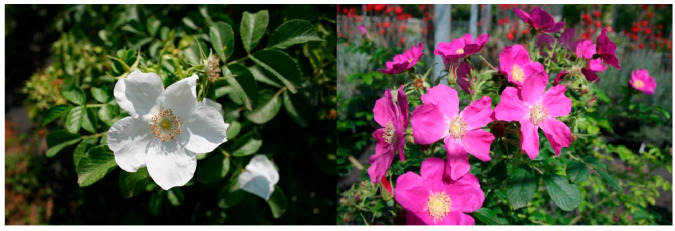
Flowers of *Rosa rugosa* cv. Alba (**left**) and cv. Rubra (**right**).

**Figure 2 plants-11-01859-f002:**
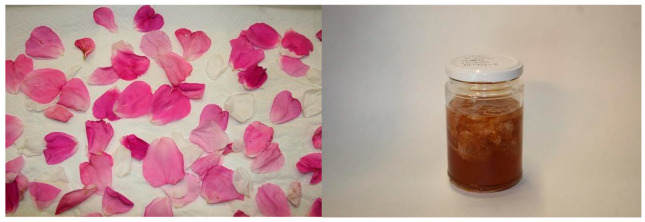
Fresh petals of Rosa rugosa flowers (**left**); maceration of cv. Rubra petals (**right**).

**Figure 3 plants-11-01859-f003:**
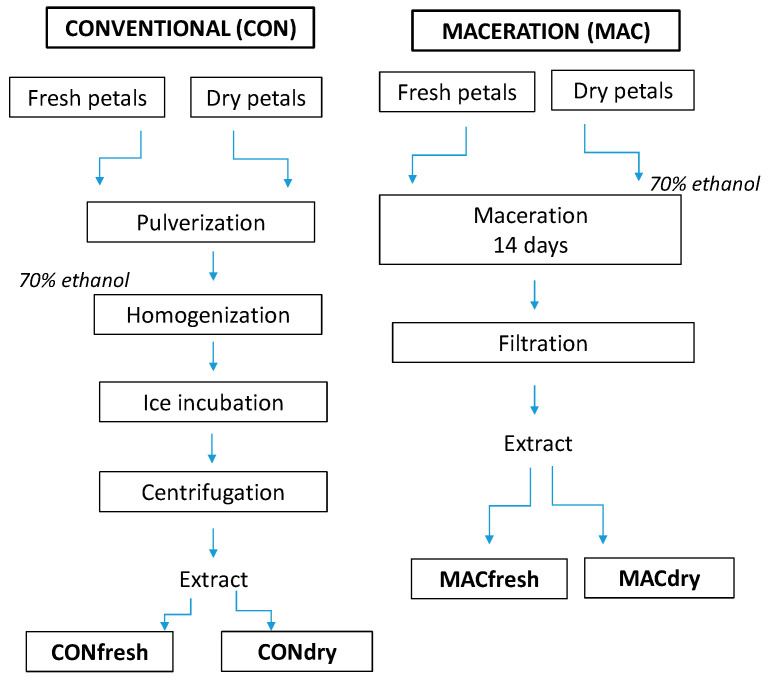
Diagram of the technological procedures for the preparation of liqueurs using maceration and conventional extraction of Rosa rugosa fresh and dry petals.

**Figure 4 plants-11-01859-f004:**
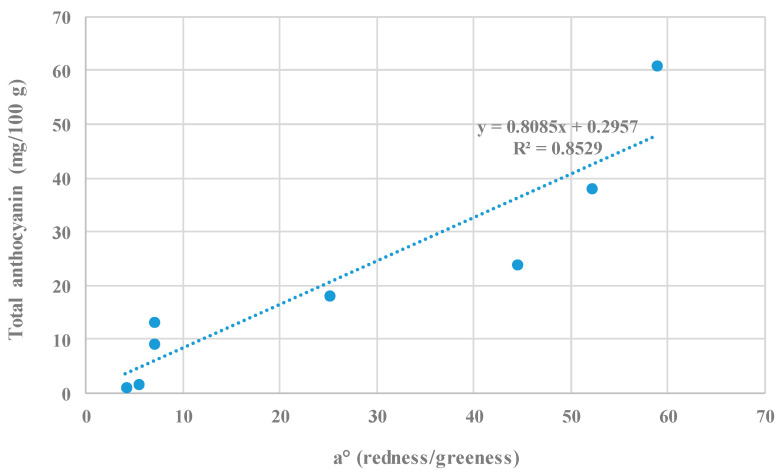
Correlation between color coordinate a° and total anthocyanin content of fresh and dry petals from *Rosa rugosa* cv. Alba and cv. Rubra.

**Figure 5 plants-11-01859-f005:**
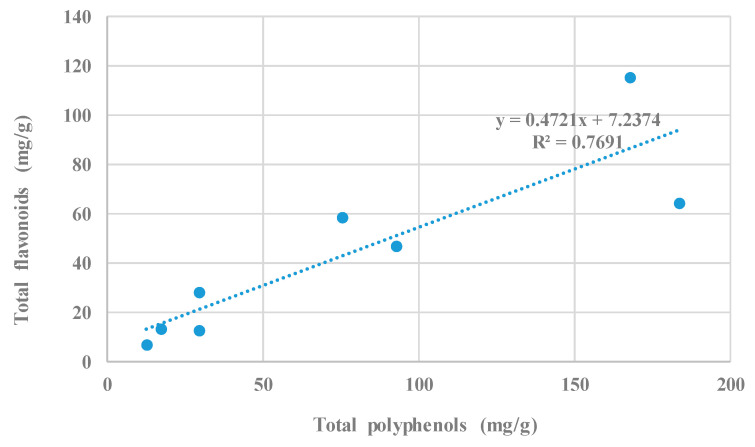
Correlation between total polyphenol and total flavonoid content of alcoholic liqueurs prepared with fresh and dry petals of *Rosa rugosa* cv. Alba and cv. Rubra.

**Figure 6 plants-11-01859-f006:**
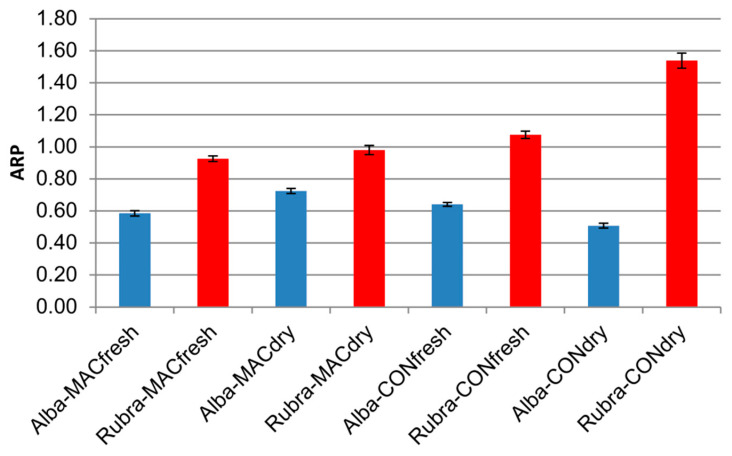
ARP values of the liqueurs from fresh and dry petals of *Rosa rugosa* cv. Alba and Rubra after maceration or conventional extraction.

**Table 1 plants-11-01859-t001:** Petal length, width, length/width ratio, fresh and dry weight, and water content of *Rosa rugosa* cv. Alba (white color) and cv. Rubra (purple color) flowers.

	Length (mm)	Width (mm)	L/W	Fresh Weight (g)	Dry Weight (g)	Water Content (%)
*Alba*	3.14 ± 0.08	3.68 ± 0.07	0.85 ± 0.01	0.19 ± 0.05	0.05 ± 0.01	73.68 ± 0.2
*Rubra*	2.83 ± 0.07	2.99 ± 0.08	0.95 ± 0.02	0.16 ± 0.03	0.03 ± 0.01	81.25 ± 0.5
Significance	*	*	*	ns	ns	*

In any column, different letters are significant at *p* ≤ 0.05 (Duncan’s test). Values are means ± standard error. ns, * = not significant and significant at *p* < 0.05 (Duncan’s test)

**Table 2 plants-11-01859-t002:** CIELab coordinates (lightness, redness/greenness, yellowness/blueness, and chroma), and hue angle of fresh and dry petals from *Rosa rugosa* cv. Alba and cv. Rubra.

	L°	*a*°	*b*°	C°	h°
*Alba* fresh	89.08 ± 0.22	5.35 ± 0.27	1.50 ± 0.09	5.59 ± 0.26	16.38 ± 1.26
*Rubra* fresh	44.58 ± 0.76	58.76 ± 0.70	−12.21 ± 0.36	60.05 ± 0.68	348.52 ± 0.42
*Alba* dry	78.48 ± 0.69	7.06 ± 0.39	36.37 ± 0.86	37.15 ± 0.87	79.10 ± 0.42
*Rubra* dry	47.92 ± 3.07	38.33 ± 4.96	−1.62 ± 0.51	44.55 ± 2.08	357.65 ± 0.73
Significance					
Cultivar (Cv)	**	**	*	**	***
Water content (Wc)	ns	*	**	ns	**
Cv * Wc	ns	ns	ns	ns	*

In any column, different letters are significant at *p* ≤ 0.05 (Duncan’s test). Values are means ± standard error. Ns, *, **, *** = not significant, significant at *p* < 0.05, 0.01, and 0.001 (Duncan’s test).

**Table 3 plants-11-01859-t003:** Total carotenoid (mg/g) and anthocyanin (mg/100 g) content in fresh and dry petals from *Rosa rugosa* cv. Alba and cv. Rubra.

	Fresh Petals		Dry Petals	
	Carotenoids	Anthocyanins	Carotenoids	Anthocyanins
Alba	0.61 ± 0.052 b	1.22 ± 0.03 b	2.14 ± 0.47 b	9.34 ± 0.02 b
Rubra	1.04 ± 0.048 a	24.05 ± 0.01 a	4.41 ± 0.06 a	60.81 ± 0.01 a
Significance	*	***	*	***

In any column, different letters are significant at *p* ≤ 0.05 (Duncan’s test). Values are means ± standard error. *, *** = not significant, significant at *p* < 0.001 (Duncan’s test).

**Table 4 plants-11-01859-t004:** Total polyphenols, total flavonoids, total anthocyanins, and selected anthocyanins content in rose liqueurs from maceration or conventional extraction of fresh and dry petals of *Rosa rugosa* cv. Alba and Rubra (expressed as mg/g of fresh or dry material). ND = Not detected.

	Fresh Petals		Dry Petals	
	Conventional	Maceration	Conventional	Maceration
	Total polyphenols			
*Alba*	12.66 ± 0.24	17.35 ± 0.14	93.0 ± 1.61	75.80 ± 1.34
*Rubra*	29.71 ± 0.81	29.75 ± 0.58	183.51 ± 5.62	167.76 ± 3.14
	Total flavonoids			
*Alba*	6.68 ± 0.1	13.35 ± 0.31	46.95 ± 2.32	58.28 ± 2.34
*Rubra*	12.84 ± 0.23	28.02 ± 0.45	64.1 ± 3.86	115.43 ± 6.64
	Total anthocyanins			
*Alba*	ND	ND	ND	ND
*Rubra*	1.96 ± 0.01	3.60 ± 0.02	11.62 ± 0.04	15.31 ± 0.04
	Cyanidin-3-O-glucoside			
*Alba*	ND	ND	ND	ND
*Rubra*	1.89 ± 0.01	1.65 ± 0.01	7.69 ± 0.03	7.25 ± 0.03
	Malvidin-3-O-glucoside			
*Alba*	ND	ND	ND	ND
*Rubra*	0.03 ± 0.01	0.02 ± 0.01	0.15 ± 0.01	0.08 ± 0.01

**Table 5 plants-11-01859-t005:** DPPH scavenging activity of rose liqueurs from fresh and dry petals of *Rosa rugosa* cv. Alba and Rubra after maceration or conventional extraction.

Rose Liqueur	SC_50_ (mg/mL)
*Alba MACfresh*	1.71 ± 0.05
*Rubra MACfresh*	1.08 ± 0.02
*Alba MACdry*	1.38 ± 0.03
*Rubra MACdry*	1.02 ± 0.03
*Alba* CONfresh	1.56 ± 0.03
*Rubra* CONfresh	0.93 ± 0.02
*Alba* CONdry	1.97 ± 0.06
*Rubra* CONdry	0.65 ± 0.02

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
