# Peer review of "Effect of Petal Color, Water Status, and Extraction Method on Qualitative Characteristics of *Rosa rugosa* Liqueur"

_plants, 2022, doi:10.3390/plants11141859_

Round 1

Reviewer 1 Report

The manuscript entitled ‘Effect of petal color and extraction method on qualitative characteristics of Rosa rugosa liqueur’ should be published in the Plants Journal Special Issue Plant Bioactive Compounds and Prospects for Their Use in Beverages after minor revision.

Materials and methods

2.3. Determination of petals total carotenoids content Please define the standard compound as witch the total carotenoid content was expressed.

Line 186. ’SC50’ should be ’SC50’ The same in the Results and Discussion section Lines 412., 414., 426.

Results and Discussion

Lines 108., 200., 214, 221., 223., 389., 436  ’R. rugosa’ should be ’R. rugosa’

Table 2. should be mentioned in the text.

Lines 243- 246. ‘Fresh petals of cv. Rubra showed a higher total anthocyanins content, with respect to those of cv. Alba (24.05 vs 1.22 mg/100 g, respectively); in the dry petals the  anthocyanins content was higher compared to that from the fresh ones, and the difference  between cultivars was more evident (60.81 and 9.34 mg/100 g for cv. Rubra and Alba, respectively) (Table 3).’ should be ‘Fresh petals of cv. Rubra showed a higher total anthocyanins content ….the difference  between cultivars is also very evident (60.81 and 9.34 mg/100 g for cv. Rubra and Alba, respectively) (Table 3).’ The relation 24.05 vs. 1.22 is higher than 60.81 vs. 9.34 so the expression ‘was more evident’ is confusing. Similarly lines 372-376, in the explanation of flavonoid content in liqueurs.

Author Response

Dear Guest Editors,

Thank you for giving us the opportunity to rework the paper for its publication in a Special Issue of Plants. We hope the article now meets the objectives and quality expectations of the journal. Please find the updated version of the manuscript attached. We would like to thank the reviewers for their hard work trying to improve the article. Below you can find a point-to point response to the reviewers. All reviewers’ comments and suggestions have been considered. We believe that the paper has been significantly improved after introducing the suggestions of the reviewers.

Kind regards

The corresponding author, Giancarlo Fascella

Reviewer 1

Materials and methods

2.3. Determination of petals total carotenoids content Please define the standard compound as which the total carotenoid content was expressed.

We have now defined the standard compound (β-carotene) as which the total carotenoid content was expressed (line 116).

Line 186. ’SC50’ should be ’SC50’ The same in the Results and Discussion section Lines 412., 414., 426.

We have correctly modified the writing both in the Materials and methods (line 203) and in the Discussion (lines 450 and 452).

Results and Discussion

Lines 108., 200., 214, 221., 223., 389., 436  ’R. rugosa’ should be ’R. rugosa’

Ok, done (see the modified text).

Table 2. should be mentioned in the text.

Ok, done (lines 226 and 236).

Lines 243- 246. ‘Fresh petals of cv. Rubra showed a higher total anthocyanins content, with respect to those of cv. Alba (24.05 vs 1.22 mg/100 g, respectively); in the dry petals the  anthocyanins content was higher compared to that from the fresh ones, and the difference  between cultivars was more evident (60.81 and 9.34 mg/100 g for cv. Rubra and Alba, respectively) (Table 3).’ should be ‘Fresh petals of cv. Rubra showed a higher total anthocyanins content ….the difference  between cultivars is also very evident (60.81 and 9.34 mg/100 g for cv. Rubra and Alba, respectively) (Table 3).’ The relation 24.05 vs. 1.22 is higher than 60.81 vs. 9.34 so the expression ‘was more evident’ is confusing. Similarly lines 372-376, in the explanation of flavonoid content in liqueurs.

Ok, done, we have modified both sentences as suggested (lines 271 and 411).

Reviewer 2 Report

I think the paper is well written and clearly presented, it is suitable for publication.

Author Response

Dear Guest Editors,

Thank you for giving us the opportunity to rework the paper for its publication in a Special Issue of Plants. We hope the article now meets the objectives and quality expectations of the journal. Please find the updated version of the manuscript attached. We would like to thank the reviewers for their hard work trying to improve the article. Below you can find a point-to point response to the reviewers. All reviewers’ comments and suggestions have been considered. We believe that the paper has been significantly improved after introducing the suggestions of the reviewers.

Kind regards

The corresponding author, Giancarlo Fascella

Reviewer 2

I think the paper is well written and clearly presented, it is suitable for publication.

Thank you, dear sir/madam, for your positive comments. We are glad that you like our article so much.

Reviewer 3 Report

Reviewers’ comments

Journal: Plants

Manuscript ID: plants-1825458

Title: “Effect of petal color and extraction method on qualitative char-2 acteristics of Rosa rugosa liqueur"

Author(s): Giancarlo Fascella, Francesca D’Angiolillo, Michele Massimo Mammano, Giuseppe Granata and Edoardo Na-4 poli

Reviewer:

Recommendation: Accept after minor revisions.

Comments: This article treats a potentially interesting topic. The authors investigated the morphological and phytochemical characteristics of petals of R. rugosa cv. Alba (white flowers) and Rubra (purple flowers), with possible use in the preparation of an alcoholic liqueur. The paper is well written, but some observations might improve the quality of the manuscript:

In abstract section: Authors need to provide a concluding sentence containing a clear-cut takeaway message for MDPI readers.

In addition, the figure quality needs to be improved particularly for Figures 4, 5 & 6.

Author Response

Dear Guest Editors,

Thank you for giving us the opportunity to rework the paper for its publication in a Special Issue of Plants. We hope the article now meets the objectives and quality expectations of the journal. Please find the updated version of the manuscript attached. We would like to thank the reviewers for their hard work trying to improve the article. Below you can find a point-to point response to the reviewers. All reviewers’ comments and suggestions have been considered. We believe that the paper has been significantly improved after introducing the suggestions of the reviewers.

Kind regards

The corresponding author, Giancarlo Fascella

Reviewer 3

Recommendation: Accept after minor revisions.

Comments: This article treats a potentially interesting topic. The authors investigated the morphological and phytochemical characteristics of petals of R. rugosa cv. Alba (white flowers) and Rubra (purple flowers), with possible use in the preparation of an alcoholic liqueur. The paper is well written, but some observations might improve the quality of the manuscript:

In abstract section: Authors need to provide a concluding sentence containing a clear-cut takeaway message for MDPI readers.

We have now added a conclusive sentence in the abstract (lines 24-25).

In addition, the figure quality needs to be improved particularly for Figures 4, 5 & 6.

We did our best to improve the figure’s quality.